# Risk Factors for Mortality of Hospitalized Adult Patients with COVID-19 Pneumonia: A Two-Year Cohort Study in a Private Tertiary Care Center in Mexico

**DOI:** 10.3390/ijerph20054450

**Published:** 2023-03-02

**Authors:** Carlos Axel López-Pérez, Francisco J. Santa Cruz-Pavlovich, Juan Eduardo Montiel-Cortés, Adriana Núñez-Muratalla, Ruth Bibani Morán-González, Ricardo Villanueva-Gaona, Xochitl Franco-Mojica, Denisse Gabriela Moreno-Sandoval, Joselyn Anacaren González-Bañuelos, Alan Ulises López-Pérez, Marily Flores-González, Cristina Grijalva-Ruiz, Edna Daniela Valdez-Mendoza, Luis Renee González-Lucano, Martín López-Zendejas

**Affiliations:** 1Escuela de Medicina y Ciencias de la Salud, Tecnologico de Monterrey, Zapopan 45201, Mexico; 2Departamento de Medicina Interna, Hospital San Javier, Guadalajara 44670, Mexico; 3Centro Universitario de Ciencias de la Salud (CUCS), Universidad de Guadalajara, Zapopan 44340, Mexico; 4Facultad de Medicina, Universidad Autónoma de Guadalajara, Zapopan 45129, Mexico

**Keywords:** COVID-19, cohort, mortality, Mexico

## Abstract

During the COVID-19 pandemic, the high prevalence of comorbidities and the disparities between the public and private health subsystems in Mexico substantially contributed to the severe impact of the disease. The objective of this study was to evaluate and compare the risk factors at admission for in-hospital mortality of patients with COVID-19. A 2-year retrospective cohort study of hospitalized adult patients with COVID-19 pneumonia was conducted at a private tertiary care center. The study population consisted of 1258 patients with a median age of 56 ± 16.5 years, of whom 1093 recovered (86.8%) and 165 died (13.1%). In the univariate analysis, older age (*p* < 0.001), comorbidities such as hypertension (*p* < 0.001) and diabetes (*p* < 0.001), signs and symptoms of respiratory distress, and markers of acute inflammatory response were significantly more frequent in non-survivors. The multivariate analysis showed that older age (*p* < 0.001), the presence of cyanosis (*p* = 0.005), and previous myocardial infarction (*p* = 0.032) were independent predictors of mortality. In the studied cohort, the risk factors present at admission associated with increased mortality were older age, cyanosis, and a previous myocardial infarction, which can be used as valuable predictors for patients’ outcomes. To our knowledge, this is the first study analyzing predictors of mortality in COVID-19 patients attended in a private tertiary hospital in Mexico.

## 1. Introduction

Two years after being declared a global pandemic by the World Health Organization (WHO) on 11 March 2020, the coronavirus disease-2019 (COVID-19), has caused more than 449,000,000 cases and 6.6 million deaths [1,2]. COVID-19, caused by the severe acute respiratory syndrome coronavirus-2 (SARS-CoV-2), is transmitted primarily through large respiratory droplets. This disease presents with a wide array of clinical presentations, ranging from asymptomatic, mild respiratory, or extrapulmonary disease, to life-threatening respiratory failure, multi-organic failure, and death [1,3,4].

Due to the magnitude of the pandemic and the current absence of an effective curative treatment, several studies have reported the clinical and epidemiological characteristics of their respective populations [1,3,4,5,6,7,8,9,10,11,12,13,14,15,16,17,18,19,20,21,22,23,24,25,26,27]. These can be used as a proxy for the prediction of patients’ outcomes. Currently, risk factors related to worse clinical outcomes and mortality include older age; male sex; obesity; comorbidities such as diabetes, hypertension, and heart failure; and laboratory features compatible with an inflammatory state [1,2,3,4,7,8,9,12,14,21,26,28,29].

Latin America and the Caribbean (LAC) has arguably been one of the areas most impacted by the pandemic, five of the region’s countries being among the 20 with the highest number of reported cases and deaths [30,31]. The pandemic has had a very elevated socioeconomic impact on the region, particularly affecting vulnerable populations: groups with a high poverty index or a lack of formal employment [21,31] as well as those with preexisting comorbidities, exacerbated by deficiencies of the health institutions in vulnerable countries [32], with most of these regions being unable to guarantee public healthcare to a considerable percentage of the population. As a response to the lack of complete public coverage, health systems in countries such as Mexico are forced to rely heavily on private spending [32,33,34].

The country has experienced six waves of the disease, resulting in more than 7.2 million cases and 331,407 deaths to date [35]. Despite not having the highest mortality rate of LAC, Mexico currently stands as the fifth country with the most deaths worldwide [35]. The alarming mortality, correlated with the aforementioned risk factors [21,23,36,37], can also be associated with the differences among healthcare institutions. Evidence suggests that the lack of homogeneity among available resources, infrastructure, quality of care, and standardized protocols may have resulted in a higher probability of dying from COVID-19 in public healthcare facilities than in private institutions [21,38,39,40]. Considering this, it is necessary to analyze the statistical behavior of the pandemic in public and private institutions independently. This would in turn present us with an image depicting the interaction between the pandemic and the two different healthcare environments, correlating with socioeconomic implications such as inequalities in healthcare access and cultural disparities of marginalized groups, which continue to impact the evolution of the pandemic in Mexico.

In this study, the findings from a 2-year retrospective large cohort study from a private tertiary care center in Guadalajara, Mexico, are reported. This study aims to describe and compare clinical characteristics, laboratory and radiological findings, and mortality among adult patients hospitalized with COVID-19 pneumonia in a Mexican private tertiary care center from April, 2020 to March, 2022.

## 2. Materials and Methods

### 2.1. Study Design

A retrospective cohort study was conducted at San Javier Hospital (SJH), a private tertiary care center located in Guadalajara, Jalisco, Mexico, that included all adult patients admitted to the hospital with a confirmed diagnosis of COVID-19 from 4 April, 2020 to 3 March, 2022. Patient admission was based on the National Institutes of Health (NIH) severity of illness categories [41], admitting all those with COVID-19 with severe or critical illness and those with moderate illness at high risk of progressing to severe disease, as determined by each attending doctor.

The primary outcome was in-hospital mortality without a set timeframe for it to occur. Inclusion criteria were: (1) adult age (≥18 years old), (2) patient admitted to SJH with a new diagnosis of COVID-19 pneumonia, (3) SARS-CoV-2 infection confirmed with RT-PCR of nasopharyngeal swab with the Berlin protocol, and (4) definite discharge or COVID-19-related death outcome. Exclusion criteria were: (1) interhospital transfer from our institution to another hospital and (2) patient discharged against medical advice. The present research was conducted in accordance with the Declaration of Helsinki, as we adhered to the General Principles of the World Medical Association; the importance of the objective outweighed the risks to the participants of the study, our research used accepted scientific principles based on the scientific literature, and all data was maintained with confidentiality [42]. The study was approved by the research ethics committee of the SJH with the register number 002-08-2022-MLZ. Due to the observational and retrospective nature of the study, no informed consent was required. Decisions regarding diagnostic approach, treatment, and follow-up were the responsibility of the attending physician, with consideration that during the pandemic different medical treatments were used based on the best scientific information available at each moment.

### 2.2. Data Collection

Epidemiological data were retrieved from the electronic medical record (TASY) of the primary and secondary evaluations performed by first-contact physicians at the respiratory care unit. Additional clinical and laboratory information, clinical outcome (survival or mortality), and pathway to death was obtained from the electronic medical record (EMR). Initial laboratory tests were defined as the first results available, typically within 24 h of hospital admission [24], including complete blood count, liver panel, basic metabolic panel, C-reactive protein (CRP), D-dimer, and Troponin I, among others.

### 2.3. Definitions

Co-morbidities were defined as follows: chronic obstructive pulmonary disease (COPD) as a diagnosis of postbronchodilator FEV1/FVC ratio of <0.70 [43]; asthma as established by the Global Initiative for Asthma 2020 [44]; chronic kidney disease (CKD) as a glomerular filtration rate below 60 mL/min for more than three months [45]; diabetes according to the guidelines of the American Diabetes Association [46]; hypertension as systolic blood pressure ≥140 mmHg and/or diastolic blood pressure ≥90 mmHg [47]; and immunosuppression as neutropenia (less than 500 neutrophils), with active malignant disease, asplenia, or under immunosuppressive treatment (prednisone >20 mg/day or other immunosuppressive drugs for at least 30 days) [21,22].

Definitions for the causes of death include: acute respiratory distress syndrome (ARDS) according to the Berlin definition [21,25], septic shock according to the 2016 Third International Consensus Definition for Sepsis and Septic Shock [48], and myocardial infarction following the guidelines of the Fourth Universal Definition of Myocardial Infarction [49].

### 2.4. Statistical Analysis and Tools

According to their distribution and type, the variables are summarized as mean and standard deviation or median with ranges and percentages (%), as appropriate. Demographic and clinical characteristics were compared between survivors versus non-survivors using a chi-square test and a t-Student test, as appropriate. Variables that proved to be statistically significant in the univariate analysis underwent multivariate ANOVA to discriminate confounding variables. The variables that remained significant with this analysis were assessed by Cox’s regression analysis method: forward likelihood ratio. We considered a two-tailed *p* < 0.05 as statistically significant.

The statistical software used for the analysis was SPSS 24.0 (SPSS Inc. Chicago, IL, USA). Figure 1 was created with Microsoft Excel version 2301 (Microsoft, Redmond, WA, USA). Appendix A was created with GraphPad Prism v.6 (GraphPad, Boston, MA, USA).

## 3. Results

In the study period, spanning from 4 April, 2020 to 3 March, 2022, 1377 patients were admitted under the diagnosis of confirmed SARS-CoV-2 pneumonia, 119 of which were excluded due to interhospital transfer or voluntary discharge against medical advice. The study population consisted of 1258 patients, of whom 1093 recovered (86.8%) and 165 died (13.1%). The median age was 56.2 ± 16.5 years, being 68.3 ± 14.2 years for non-survivors and 54.4 ± 16.0 for survivors. The mean length of stay was 12.2 ± 13.7 days, being significantly higher in those patients who died compared with survivors. In total, 243 (19.3%) of patients were admitted to the intensive care unit (ICU), and 200 (15.9%) were mechanically ventilated (MV). A significant association was observed between the need for MV or ICU admission and in-hospital death. Among survivors, 86 (7.8%) received mechanical ventilation, and 107 (9.7%) were managed in the ICU. Figure 1 shows patient distribution regarding the number of hospital admissions, hospital discharges, ICU admissions, and in-hospital deaths during the study period. Similar to those observed in the general population, three waves of disease are denoted in the figure during the study period, reaching the peak of hospital admissions in December 2020, August 2021, and January 2022.

Demographic, clinical, and laboratory characteristics at admission of survivors and non-survivors are shown in Table 1, Table 2 and Table 3. Several of these variables showed statistical significance (*p* < 0.05) in univariate analysis.

The mechanisms for death are summarized in Appendix A. The most common cause was multi-organic failure (42.4%), followed by ARDS (33.9%) and septic shock (10.9%). Other causes included unstable bradycardia, pulmonary embolism, myocardial infarction, and hypovolemic shock, which were much less common. Vaccination status in both survivors and non-survivors can be seen in Appendix A. Observed and expected values of the variables that were analyzed using the chi-square test can be found in Appendix A. Nonparametric plots comparing MULBSTA, Charlson, and NEWS scales scores between survivors and non-survivors can be found in Appendix A.

In the multivariate analysis (Table 4), the variables that independently predicted mortality, identified by Cox regression analysis, were older age (>60 yo), cyanosis, and previous myocardial infarction.

## 4. Discussion

To our knowledge, this is the first large cohort study of COVID-19 in-hospital mortality and the associated risk factors of patients attended exclusively in a private hospital in Mexico, and one of the few in LAC. In 2021, LAC was the region with the highest number of COVID-19 deaths and deaths per 1000 population, representing 28.8% of global reported deaths while having only 8.4% of its population [50]. In our cohort, in-hospital overall mortality was 13.1%, which contrasts with the mortality reported by other hospitals in this country (22–53%) [21,22,23,25,26] as well as with some cohorts in other LAC countries [5,18].

The significantly lower mortality rate found in our cohort can be explained by several factors, namely, the fact that our hospital belongs to the private health subsystem compared with other Mexican cohorts based in public health services [21,22,23,25,26]. Márquez-González et al. [27], Carrillo-Vega et al. [51], and Salinas-Escudero et al. [52] analyzed the national database to identify the risk factors for hospitalization and death in the Mexican population, showing a lower patient survival rate among those hospitalized in public institutions. This problem is prevalent among health systems in most LAC countries, which, to varying degrees, all lack universal public health regimes, instead relying heavily on private subsystems and, in most cases, considerable out-of-pocket expenses [32,53,54,55]. In their cohort study of a private healthcare network in Brazil, De Oliveira et al. also reported a considerably lower mortality rate compared with other cohorts from the public subsystem in Brazil and other parts of the world [5]. Aside from age, which was also lower than the reported mean in other studies, they attributed the disparities between private and public hospitals to be a possible factor involved in this difference. The Mexican health system’s highly heterogeneous organization and quality of care have allowed discrepancies in healthcare to persist to date. The system of care is divided into four main subsystems (private healthcare providers and the public institutions Instituto Mexicano del Seguro Social (IMSS), Instituto de Seguridad y Servicios Sociales para los Trabajadores del Estado (ISSSTE), and Secretaría de Salud (SS)), all of which remain fragmented and incapable of delivering universal care [32,33,34,38,56]. Public institutions represent the health services with the highest demand, which puts them at a higher risk of exceeding their operating capacity—resulting in hospital saturation and heightened mortality [51].

Another factor to consider for the difference in mortality rates is that, while many Mexican cohorts analyzed the first months of the pandemic, our study spanned a 2-year period. Thus, the evolution of our clinical knowledge of COVID-19, a lesser degree of bed-saturation and overcrowding of critical areas, and the effect of vaccines over the last months of our studied period, most likely contributed to a decrease in in-hospital mortality. On the other hand, the inclusion of patients with an initial moderate NIH severity of illness probably contributed to this result, although only 90 patients with this characteristic were present in the study population. Finally, a factor that was not considered in our study was the effect of the newly developing COVID-19 variants. One of particular relevance is the Omicron variant reported in November 2021, the fifth variant of concern (VOC) posing a threat to global public health. Omicron emerged as the variant most mutated, transmissible, and resistant to immunotherapeutics and vaccines. Nonetheless, Omicron proved to be milder than the previous variants, mostly causing upper respiratory tract symptoms and resulting in low mortality rates [57,58].

In our study, 19.3% of patients received care in the ICU and 15.9% were MV. ICU and MV mortality were 55.9% and 57%, respectively, similar to other Mexican [21,23,24,25] and global [5,20] cohorts. Both ICU admission and the need for MV were significantly more frequent in non-survivors, which has been commonly reported amongst many cohorts, highlighting the importance of ICU management and MV as predictors of death in patients hospitalized due to COVID-19 [5,20,25,26,27].

Hypertension and diabetes are comorbidities identified by several studies as risk factors for mortality. Although they were identified as predictive in the univariate analysis, they were not included in the final multivariate model. They both present in a high prevalence in LAC and the Mexican population [21,25,59]. Hypertension is one of the comorbidities that has most commonly been associated with increased mortality in COVID-19 patients, though the exact mechanism remains unclear [10,11,22,25,59,60,61]. Its prevalence in our cohort was similar to the national average (31%) and to that of other LAC countries [4,21,25]. The use of ACEI/ARBS represented a significant difference between both groups. Although mediated by a possible mechanism by which RAAS blockers increase ACE2 expression, potentially increasing the risk of SARS-CoV-2 infection, the effect of ARB or ACEI use on disease severity is still controversial [22,25]. In our cohort, diabetes presented with a higher prevalence than the national average (13.7%) [21]. As with hypertension, it has been associated with COVID-19 severity and mortality [3,19,59,61,62], with many proposed mechanisms, including reduced resistance to viral infections as a consequence of a sustained low level of immunity as well as vascular and heart damage due to longstanding disease [62].

Overweight status and obesity showed no difference between the two groups. Though it has been associated with increased disease severity in COVID-19 patients in some studies, the association remains unclear, with mixed results among the bibliography [5,18,25]. A meta-analysis conducted by Mesas et al. [61] showed that increased mortality was present only in studies with fewer chronic or critical patients, by which BMI did serve as a prominent prognostic factor only in studies with these conditions, which was not the case in our study.

Immunosuppression [3,5,12,15,62,63], cancer [10,59,61,62], and chronic kidney disease [12,17,20,52,61] are other important comorbidities that have been reported as predictive risk factors for mortality in different cohorts. Despite the fact that they were significantly more frequent in the mortality group in our cohort, they did not remain significant in the final multivariate model.

After the univariate analysis, significant variables were analyzed by multivariate ANOVA and then by Cox regression analysis to determine the explicative and predictive variables. In the resulting model, older age, the presence of cyanosis, and previous myocardial infarction were the main predictors of mortality, consistent with the findings amongst other cohorts. In several studies, age was found to be a main determinant of COVID-19-related in-hospital mortality, independent of other pre-existing comorbidities [64,65,66,67]. The median age in our study was 56.2 ± 16.5, similar to other large cohorts in our country [21,22,23,25,26]. As previously established, age has been reported as one of the most important risk factors, being associated with higher mortality plus extended hospital and ICU times [6,27,59]. In our study, age was identified as a risk factor for mortality (non-survivors, were, on average, 14 years older than survivors) and remained as an independent mortality risk factor after multivariate analysis. This may be explained by contributing factors such as age-related physiological changes, impaired immune function, and preexisting illnesses [18,20,59,62]. At this point in the pandemic, older age is well established as a strong predictor of severity and mortality in patients with COVID-19, which prompts early referral of older individuals for inpatient care [11,19,28,60]. In one study conducted in the same city as our present research, age, along with other factors, was also found to be a mortality predictor in multivariate analysis [25]. Another predictive variable was the presence of cyanosis. Although identified as a mortality-related risk factor in the univariate analysis of some studies, our cohort, to the best of our knowledge, is the first to include it in the final multivariate model [18,68]. Finally, the history of previous myocardial infarction was also an important predictor of mortality. The presence of cardiovascular disease has been extensively reported with worse outcomes in patients with COVID-19 [3,8,12,61,69]. Specifically, a history of ischemic heart disease was found to be a significant variable by some cohorts [3,20]. Similar to our results, one study also reported myocardial infarction as a predictor of mortality in the multivariate analysis [67].

An important aspect to consider while analyzing COVID-19 mortality is the evaluation of the role of SARS-CoV-2 infection in such deaths [70]. At the start of the pandemic, some COVID-19 deaths may have been misclassified as being due to other causes, while conversely, during peak pandemic periods, a bias in the opposite direction probably occurred [19]. Due to the lack of knowledge of the pathophysiology of COVID-19 death, as well as the high prevalence of comorbidities observed in deceased people who tested positive for SARS-CoV-2, the question of whether a patient died with or due to COVID-19 is still very much debated [71,72]. Assigning a primary cause of death to a deceased patient with multiple principal diagnoses that could lead to death has been challenging since before COVID-19 [73]. The problem of objectively identifying the “real” cause of death is not only relevant from a conceptual standpoint, but also has many practical consequences regarding epidemiology, public health interventions and policies, health communication to the public, and political decisions [74]. Although numerous observational studies have reported outcomes and risk factors for mortality in COVID-19, the accuracy of the causes of death has seldom been reported [75]. This can be due to many factors, including the methods of assigning primary cause of death, the impossibility of performing necropsies, and countries’ laws allowing only one cause to be reported on a death certificate [70,71,73,74,76]

Regarding the limitations of this study, its retrospective nature makes it prone to under-documentation of many clinical variables, limiting the researchers’ capacity to obtain comprehensive data due to incomplete medical records. This was particularly relevant for determining the actual role of SARS-CoV-2 infection in each death, as the EMR often lacked the information necessary to evaluate whether COVID-19 was only an epiphenomenon for that particular death. Social determinants of the study population, such as median household income, were not assessed. As genomic sequencing data was not available, analysis of the predominant variants of concern in each wave could not be performed. Due to the changing nature of the pandemic, along with the growing understanding of the disease, clinical practice improvements were implemented, with the evaluation of such changes exceeding the scope of this study [5]. Finally, we excluded patients that did not have the entire course of disease in our institution, such as those discharged against medical advice or because of interhospital transfer, as we were therefore unable to assess their evolution. Despite these limitations, the size and duration of this study allowed us to provide a reasonably complete overview of the pandemic as it presented in our hospital [77].

Our study gains relevance as the socioeconomic impact of COVID-19 continues to impact the population of our country, worsening socioeconomic inequality: while nonvulnerable groups are given the option of more reliable services, the more marginalized populations are left with no choice but to attempt to receive care in saturated, underfunded, and often uncoordinated public health subsystems [78]. These disparities further heighten inequalities affecting vulnerable groups, including indigenous communities, migrants, people in overcrowded living conditions, informal workers, people with disabilities, and older adults, even more so in cases involving chronic diseases, which are also correlated with these same vulnerabilities [21,30,37,50,54,55,78]. While this is not limited to Mexico or LAC—the syndemic relationship among social inequalities, chronic diseases, and COVID-19 has been reported at an international level [79]—the public and private subsystems’ conditions, low healthcare spending, infrastructure, and other health-related policies have all had a considerably higher socioeconomic impact in LAC [32].

## 5. Conclusions

Mortality in hospitalized patients with COVID-19 in this Mexican private tertiary care center was 13.1%. Older age, the presence of cyanosis, and a previous myocardial infarction were the most significant independent risk factors for mortality in adult patients hospitalized with COVID-19 pneumonia in our 2-year cohort. Considering the significant disparities in the quality of care that exist between the private and public health subsystems in Mexico, our results gain special significance, as they contribute to a more complete overview of the healthcare system and its interaction with the pandemic.

## Figures and Tables

**Figure 1 ijerph-20-04450-f001:**
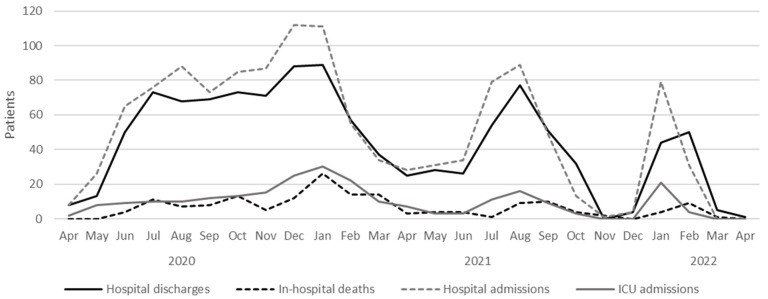
Number of hospital discharges, in-hospital deaths, hospital admissions, and ICU admissions from 4 April 2020 to 3 March 2022.

**Table 1 ijerph-20-04450-t001:** Comparisons of the demographic and clinical characteristics of survivors and non-survivors at admission.

Variable	Total n (%)	Survivor n (%)	Non-Survivor n (%)	*p* Value ^1^
Sex (Male/Female), n	845/413	741/352	104/61	0.248 *
Age ($\bar{x}$ ± S.D.) years	56.2 ± 16.5	54.4 ± 16.0	68.3 ± 14.2	<0.001 ^#^
BMI classification, %				
Underweight	10 (0.8%)	9 (0.8%)	1 (0.6%)	
Normal BMI	240 (19.1%)	199 (18.4%)	41 (24.8%)	
Overweight status	486 (38.6%)	423 (38.3%)	63 (38.2%)	0.399 *
Obesity grade I	322 (25.6%)	285 (26.1%)	37 (22.4%)	
Obesity grade II	114 (9.1%)	103 (9.3%)	11 (6.7%)	
Obesity grade III	86 (6.8%)	74 (7.0%)	12 (7.3%)	
Hypertension (%)	427 (33.9%)	346 (31.7%)	81 (49.1%)	<0.001 *
Diabetes (%)	270 (21.5%)	216 (19.8%)	54 (32.7%)	<0.001 *
COPD (%)	34 (2.7%)	22 (2.0%)	12 (7.3%)	0.001 *
Asthma (%)	27 (2.1%)	25 (2.3%)	2 (1.2%)	0.422 *
Immunosuppression (%)	56 (4.5%)	42 (3.8%)	14 (8.5%)	0.010 *
Cancer (%)	60 (4.8%)	42 (3.8%)	18 (10.9%)	<0.001 *
HIV (%)	2 (0.2%)	2 (0.2%)	0 (0%)	>0.999 *
Previous stroke (%)	16 (1.3%)	11 (1.0%)	5 (3.0%)	>0.999 *
CKD (%)	49 (3.9%)	39 (3.6%)	10 (6.1%)	0.021 *
Organ transplant recipient (%)	16 (1.3%)	14 (1.3%)	2 (1.2%)	>0.999 *
Chronic liver disease (%)	9 (0.7%)	8 (0.7%)	1 (0.6%)	>0.999 *
ACEI/ARA (%)	283 (22.5%)	234 (21.4%)	49 (29.7%)	0.021 *
Previous myocardial infarction (%)	35 (2.8%)	23 (2.1%)	12 (7.3%)	0.001 *
Depression (%)	17 (1.4%)	15 (1.4%)	2 (1.2%)	0.830 *
Smoker (%)				
Never	808 (64.2%)	715 (65.4%)	93 (56.4%)	
Unknown	301 (23.9%)	257 (23.5%)	44 (26.7%)	0.071 *
Currently	65 (5.2%)	54 (4.9%)	11 (6.7%)	
Former	84 (6.7%)	67 (6.1%)	17 (10.3%)	
Alcohol use (%)				
Never	805 (64.0%)	700 (64.0%)	105 (63.6%)	
Unknown	330 (26.2%)	279 (25.5%)	51 (30.9%)	0.114 *
Currently	109 (8.7%)	102 (9.3%)	7 (4.2%)	
Former	14 (1.1%)	12 (1.1%)	2 (1.2%)	
COVID-19 pneumonia on imaging (%)	1133 (90.1%)	981 (90.2)	152 (92.1)	0.480 *
NIH severity scale (%)				
Critical	64 (5.1%)	26 (2.4%)	38 (23.0%)	
Moderate	90 (7.2%)	85 (7.8%)	5 (3.0%)	<0.001 *
Severe	1104 (87.8%)	982 (89.8%)	122 (73.9%)	
MuLBSTA ($\bar{x}$ ± S.D.)	6.3 ± 3.3	6.0 ± 3.2	8.3 ± 3.6	<0.001 ^#^
Charlson ($\bar{x}$ ± S.D.)	2.1 ± 2.9	1.8 ± 3.0	3.5 ± 2.1	<0.001 ^#^
qSOFA (%) n				
0	354 (28.1%)	329 (30.1%)	25 (15.2%)	
1	816 (64.9%)	714 (65.3%)	102 (61.8%)	<0.001 *
2	71 (5.6%)	47 (4.3%)	24 (14.5%)	
3	17 (1.4%)	3 (0.3%)	14 (8.5%)	
NEWS ($\bar{x}$ ± S.D.)	6.3 ± 2.5	6.1 ± 2.2	7.9 ± 3.1	<0.001 ^#^

^1^ The comparisons are between survivors and non-survivors; * chi-square test with Exact Fisher’s test if apply; ^#^ Student’s *t* test.

**Table 2 ijerph-20-04450-t002:** Comparisons of signs and symptoms between non-survivors and survivors at admission.

Variable	Total n (%)	Survivor n (%)	Non-survivor n (%)	*p* Value ^1^
Days onset symptom-admission ($\bar{x}$ ± S.D.)	9.2 ± 5.2	9.1 ± 5.1	10.1 ± 6.1	0.027 ^#^
Length of stay, days ($\bar{x}$ ± S.D.)	12.2 ± 13.7	10.6 ± 10.4	22.9 ± 24.5	<0.001 ^#^
Heart rate ($\bar{x}$ ± S.D.)	89.0 ± 19.2	88.6 ± 18.1	91.8 ± 24.9	0.043 ^#^
Respiratory rate ($\bar{x}$ ± S.D.)	25.4 ± 7.3	25.2 ± 7.0	27.5 ± 8.7	<0.001 ^#^
% Saturation O_2_ ($\bar{x}$ ± S.D.)	82.1 ± 11.6	83.4 ± 10.2	74.0 ± 15.9	<0.001 ^#^
Fever (%) n	756 (60.1%)	658 (60.2%)	98 (59.4%)	0.865 *
AVPU score				
Alert	1196 (95.1%)	1066 (97.5%)	130 (78.8%)	
Voice	30 (2.4%)	17 (1.6%)	13 (7.9%)	<0.001 *
Pain	7 (0.6%)	1 (0.1%)	6 (3.6%)	
Unresponsive	25 (2.0%)	9 (0.8%)	16 (9.7%)	
Cough (%)	838 (66.6%)	726 (66.4%)	112 (67.9%)	0.724 *
Headache (%)	538 (42.8%)	478 (43.7%)	60 (36.4%)	0.077 *
Dyspnea (%)	1041 (82.8%)	892 (81.6%)	149 (90.3%)	0.008 *
Diarrhea (%)	252 (20.0%)	224 (20.5%)	28 (17.0%)	0.300 *
Chest pain (%)	224 (17.8%)	193 (17.7%)	31 (18.8%)	0.743 *
Chills (%)	309 (24.6%)	275 (25.7%)	34 (16.4%)	0.210 *
Odynophagia (%)	308 (24.5%)	281 (37.4)	27 (29.7%)	0.011 *
Myalgias (%)	458 (36.4%)	409 (37.4%)	49 (29.7%)	0.057 *
Arthralgias (%)	406 (32.3%)	357 (32.7%)	49 (29.7%)	0.476 *
Malaise (%)	880 (70.0%)	769 (70.4%)	111 (67.3%)	0.466 *
Rhinorrhea (%)	151 (12.0%)	135 (12.4%)	16 (9.7%)	0.370 *
Vomiting (%)	93 (7.4%)	83 (7.6%)	10 (6.1%)	0.529 *
Abdominal pain (%)	77 (6.1%)	64 (5.9%)	13 (7.9%)	0.382 *
Conjunctivitis (%)	22 (1.7%)	20 (1.8%)	2 (1.2%)	0.757 *
Cyanosis (%)	101 (8.0%)	71 (6.5%)	30 (18.2%)	<0.001 *
Anosmia (%)	145 (11.5%)	135 (12.4%)	10 (6.1%)	0.018 *
Dysgeusia (%)	129 (10.3%)	118 (10.8%)	11 (6.7%)	0.129 *
Glasgow Coma Scale <15 (%)	77 (6.1%)	43 (3.9%)	34 (20.6)	<0.001 *

^1^ The comparisons are between survivors and non-survivors; * chi-square test with Exact Fisher’s test if apply; ^#^ Student’s *t* test.

**Table 3 ijerph-20-04450-t003:** Comparisons of laboratory findings between non-survivors and survivors at admission.

Variable	Total	Survivor	Non-Survivor	*p* Value ^1^
Urea (mg/dL) ($\bar{x}$ ± S.D.)	44.1 ± 30.1	41.8 ± 26.8	59.3 ± 43.4	<0.001 ^#^
Creatinine (mg/dL) ($\bar{x}$ ± S.D.)	1.0 ± 1.1	1.0 ± 1.2	1.1 ± 0.8	0.489 ^#^
Total bilirubin (mg/dL) ($\bar{x}$ ± S.D.)	0.7 ± 0.6	0.8 ± 0.6	0.8 ± 0.4	0.757 ^#^
AST (U/L) ($\bar{x}$ ± S.D.)	60.8 ± 57.8	60.0 ± 53.6	66.8 ± 82.3	0.242 ^#^
ALT (U/L) ($\bar{x}$ ± S.D.)	60.9 ± 67.4	62.6 ± 69.9	48.2 ± 43.2	0.033 ^#^
Alkaline phosphatase (U/L) ($\bar{x}$ ± S.D.)	102.1 ± 61.4	101.4 ± 59.8	107.2 ± 72.2	0.349 ^#^
Albumin (g/dL) ($\bar{x}$ ± S.D.)	3.6 ± 0.5	3.7 ± 0.5	3.3 ± 0.5	<0.001 ^#^
CRP (mg/L) ($\bar{x}$ ± S.D.)	132.2 ± 101.3	128.4 ± 99.5	158.9 ± 110.1	0.001 ^#^
Leukocytes (−10^3^/uL) ($\bar{x}$ ± S.D.)	9.9 ± 6.8	9.6 ± 6.8	12.0 ± 6.5	<0.001 ^#^
Hb (g/dL) ($\bar{x}$ ± S.D.)	14.4 ± 2.2	14.5 ± 2.2	13.7 ± 2.4	<0.001 ^#^
Platelets (10^3^/uL) ($\bar{x}$ ± S.D.)	431.8 ± 5674.5	456.7 ± 6092.0	269.0 ± 176.9	0.700 ^#^
Absolute lymphocyte count ($\bar{x}$ ± S.D.)	1285.1 ± 1089.6	1260.6 ± 905.3	1447.6 ± 1903.7	0.234 ^#^
Absolute neutrophil count ($\bar{x}$ ± S.D.)	7173.3 ± 3960.6	6978.7± 3832.7	8535.34 ± 547.3	<0.001 ^#^
Neutrophil-lymphocyte ratio ($\bar{x}$ ± S.D.)	7.3 ± 6.3	6.8 ± 5.2	10.7 ± 10.4	<0.001 ^#^
D-Dimer (ng/mL) ($\bar{x}$ ± S.D.)	1109.3 ± 2194.0	933.8 ± 1947.3	2342.0 ± 3214.6	<0.001 ^#^
CPK (U/L) ($\bar{x}$ ± S.D.)	213.0 ± 658.6	219.0 ± 699.8	179.5 ± 354.5	0.718 ^#^
LDH (U/L) ($\bar{x}$ ± S.D.)	387.9 ± 212.2	371.6 ± 183.0	501.1 ± 334.0	<0.001 ^#^
Fibrinogen (mg/dL) ($\bar{x}$ ± S.D.)	422.5 ± 170.6	414.0 ± 162.2	490.5 ± 299.1	0.566 ^#^
Troponin I (ng/mL) ($\bar{x}$ ± S.D.)	0.18 ± 3.4	0.0 ± 0.1	1.1 ± 9.0	0.291 ^#^
pH ($\bar{x}$ ± S.D.)	7.4 ± 0.1	7.4 ± 0.1	7.4 ± 0.1	0.064 ^#^
Partial pressure O2 (mmHg) ($\bar{x}$ ± S.D.)	80.4 ± 40.1	79.0 ± 39.6	82.9 ± 41.2	0.555 ^#^
Partial pressure CO2 (mmHg) ($\bar{x}$ ± S.D.)	42.4 ± 18.2	38.2 ± 12.7	49.9 ± 23.4	0.001 ^#^
Lactate (mmol/L) ($\bar{x}$ ± S.D.)	567.3 ± 5005.8	445.0 ± 4448.9	784.0 ± 5902.1	0.684 ^#^

^1^ The comparisons are between survivors and non-survivors; ^#^ Student’s *t* test.

**Table 4 ijerph-20-04450-t004:** Results of the multivariate Cox regression analysis to predict mortality in patients with COVID-19.

Variable	Exp (B)	95% CI	*p* Value
>60 years	2.445	1.679–3.561	<0.001
Cyanosis	1.825	1.195–2.787	0.005
Previous myocardial infarction	1.930	1.058–3.520	0.032

Cox regression analysis method: forward likelihood ratio. Independent variables included in the initial model: age (0 = ≤60 years old, 1 = >60 years old), urea (0 = 12−54 mg/dL, 1 = <12 mg/dL, 2 = >54 mg/dL), location of care (intensive care unit = 0, non-intensive-care unit = 1), previous myocardial infarction (no = 0, yes = 1), chronic lung disease (no = 0, yes = 1), cancer (no = 0, yes = 1), O2 saturation (0 = ≥90%, 1 = <90%), cyanosis (no = 0, yes = 1), dysgeusia (no = 0, yes = 1), D-dimer (0 = ≥250, 1 = <250), and anosmia (no = 0, yes = 1). Variables not included due to overlapping with other variables: NIH severity scale, MuLBSTA, Charlson, qSOFA, NEWS, AVPU, and Glasgow Coma Scale. Continuous variables were transformed into categorical ones in order to obtain the narrowest confidence intervals and to simplify the interpretation of our model.

## Data Availability

The datasets used and/or analyzed during the current study are available from the corresponding author on reasonable request.

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
