# Peer review of "Risk Factors for Mortality of Hospitalized Adult Patients with COVID-19 Pneumonia: A Two-Year Cohort Study in a Private Tertiary Care Center in Mexico"

_ijerph, 2023, doi:10.3390/ijerph20054450_

Round 1

Reviewer 1 Report

The manuscript with title ”Risk factors for mortality of adult patients with COVID-19: a 2 two-year cohort study in a private tertiary care center in Mexico” contributes to the ongoing Covid-19 discussion.

I am in general positive towards publication but I think the manuscript will need a major revision before I can recommend it for publication. My reasons for this are outlined in the following.

The study behind the manuscript is fairly well described and yet the formulated aim seems to reach beyond the scope of this study: “This study aimed to describe and compare clinical characteristics, laboratory and radiological findings, and mortality among adult patients hospitalized with COVID-19.” There are limitations and constraints in study context that should be addressed in the formulation of the aim. I will give a few examples. The study is not concerning itself with any type of hospitalized Covid-19 case but only those admitted to a private tertiary hospital in Mexico. There are also time constraints as well as case definition constraints. This latter constraint is only found in the beginning of the result section and should in fact be emphasized in the aim. In section 3 we can read that “In the study period, spanning from April 4th, 2020, to March 3rd, 2022, 1,377 patients were admitted under the diagnosis of confirmed SARS-CoV-2 pneumonia...”. I find this important since most people who get infected with COVID-19 have mild or moderate symptoms, however some contracting COVID-19 get pneumonia and COVID-19 pneumonia is a serious illness. I think this should also be reflected in the title and in the conclusions that this restriction of cases exist in this study.

In section 2.1 the authors claim that the study is adhering to the declaration of Helsinki and this is of course convenient. I think it is much better to declare what was actually done to adhere to this declaration or even better to really describe what was done to conform with ethical standards.

In section 2.3 the authors state that: “The cause of death was retrieved according to the data reported in the electronic medical record, as determined by the attending doctor.” This is a big issue, how were the causes of death weeded out? Do you die from Covid or with Covid. Since this is the outcome measure of this study it is extremely important to show how this was established and also to clearly report problems if they exist. At the end of this section several causes of death are mentioned but not Covid-19. I guess this is deliberate but needs an explanation.

In section 2.4 statistical analysis I find it difficult to follow the logic in this sentence. “Characteristics at admission were compared between survivors at discharge versus non-survivors using a chi-squared test or Fisher’s exact test if qualitative, and a t-student test or ANOVA if quantitative with normal distribution.” What comparisons were actually made? Why are certain tests done when something is quantitative with normal distribution, does this refer to variables in the study? I also want to mention that many statistical tests based on the assumption of a normal distribution do not take into account what distribution the variables seem to have, this pertains to other properties. This section needs a clarification and an English language polish.

The sentence “After the univariate analysis, mortality risk factors were analyzed by Cox regression analysis to determine explicative and predictive variables...” also raises questions. Nowhere in this manuscript are there any indications of what set of variables have undergone this Cox regression and nowhere we see what was considered important to adjust for. Furthermore we receive no guidance from the authors concerning how to interpret the multitude of bivariate results given what was found in the Cox regression analysis. This should, at least, be addressed in the discussion section. What i mean by this is, are all these bivariate results valid given the result of the multiple regression? Hard for a reader to know given the very small amount of information given concerning the Cox regression.

The result section is clear and concise albeit rather lengthy in the presentation with respect to the great number of bivariate comparisons. If I counted correctly there are 73 bivariate statistical comparisons made. Several show statistical significance and I wonder if the authors have considered the multiple comparisons problem? Given the large amount of bivariate statistical tests it could be wise to consider this issue and either use modern corrections or issue a statement of caution.

In the discussion section 4 it would very good with a paragraph discussing both the distinction and implications of dying with Covid-19 as compared to when the cause of death is Covid-19. I find this is a topic often omitted and in this study it could be helpful and a real strength.

Finally in the conclusions section I find the statement “To our knowledge, this is the first study analyzing predictors of mortality in COVID-19 patients attended on a private tertiary hospital in Mexico.” a bit out of place. This has already been well established in the discussion and is probably not a conclusion that can be drawn based on the results from this nice study.

Minor comments:

Table 1 mentions Asma which could be a spelling error

The authors decided to highlight all p-values deemed statistically significant in bold but this is not stringent in every table even if the result from the Cox-regression is excluded.

Reviewer 2 Report

1. In Statistical analysis Section, the authors must discuss the normality test of data?.

2. In Statistical analysis Section, the authors must discuss the homogeneity test of observations?.

3. Non-parametric plots must be sketched such as quantile-quantile, histogram, kernel density, among others for data. The authors should report the data inside the paper.

4. Did the author discuss the conditions for the multivariate Cox’s regression analysis to predict mortality in patients with COVID-19?. Discuss. 

5. Since the statistical software SPSS 24.0 has been used. What about the extreme or outliers values?. Discuss the approach which has been applied to avoid such these observations. 

6. The name of the tests should be listed inside the text for the readers.  

7. Abbreviation and statistical tools Sections should be added.

8. To use Chi-squared test, the authors compared between the observed and expected values. Please include these values inside the tables for the readers. 

Reviewer 3 Report

Major:

1. Why the mortality risk factors in cox regression analysis are only based on age, cyanosis, and history of myocardial infarction?

2. Exclusion and inclusion criteria are unclear. Authors simply neglect the vaccination history and effects of immunosuppressed or malignant conditions. 

3. Symptoms could be different based on the VOC (such as Omicron; ref. https://doi.org/10.1016/j.jiph.2022.11.024). Therefore, authors should analyze the data based on the dominating variants.

Minor:

1. Please include the age range for young adult in the method.

Round 2

Reviewer 1 Report

I find that the manuscript has improved substantially and could be a good candidate for publication. However there are a few remaining issues I think must be addressed before I can fully endorse publication. I find one main issue I would like to point out and two minor.

The biggest issue pertains to the way the analyses are presented in the result section and how these results subsequently are dealt with in the discussion.

The way the authors describe the procedure of variable selection for the final analyses clarifies how this process was conducted. The principles that guided the authors on how variables were selected for the Cox regression could be debated but since the chosen path is on the conservative side it is fine. The more pressing matter is that the author response does have implications for the large amount of bivariate analyses performed. It is a bit unfortunate that my question about the relevance of these bivariate analyses were not addressed by the authors in the previous iteration. However since this section now is recognized as more of a precursor for the Cox analyses this should be reflected in the manuscript. When confounding and the importance of a large group of variables have been properly assessed, then some of the primary bivariate relationships then loses something of their importance since they, as a consequence and in cases, could be regarded as the result of, for instance, confounding. This means that the  bivariate section in results, in the light of the author response, is surprisingly large given the little emphasis put on the main outcome which I understand to be the Cox regression part. Naturally it can be good to present what is relevant but keeping in mind this is a preparation for further analyses. The results of the bivariate analyses with respect to the chosen outcome therefore, in cases, lack the necessary weight and relevance. I suggest the authors limit this section since many of the results herein can be debated. This is also true concerning the discussion section of the manuscript. Almost a full page of text is dedicated to these bivariate findings not identified in the multiple analyses. Reconsideration by the authors seems prudent.

Minor issue include:

I find it remarkable that not a single continuous variable was allowed in its original form in this study. It is clearly reported and in that way it’s fine but from an epidemiological point of view it makes me wonder. There might exist reasoning behind the choices of the various cut-offs but such arguments are not revealed.

Simple calrification:

The answer the authors provide concerning how they select statistical methods in section 2.4 displays a bit of misunderstanding. I never asked for tests of normality, on the contrary I was arguing that the distribution of the variables under scrutiny is not what matters in this context. In the text it is the author group that makes claims about the meaning of normality. “..t-student test or ANOVA if quantitative with normal distribution.” Since this statement now is removed things are nevertheless better.

All in all the manuscript has improved substantially.

Reviewer 2 Report

All comments have been done 

Author Response

Thank you for your comments.

Reviewer 3 Report

Authors have made necessary revisions to their manuscript, which I appreciate a lot. Several suggestions prior to the publication:

1.     Subjects with immunosuppression significantly affected the data, I suggest authors to make thorough discussion about this.

2.     Also the effect of omicron should be discussed too. Authors may refer to this article: https://doi.org/10.1016/j.jiph.2022.11.024
